# Sperm Quality Assessment in Stallions: How to Choose Relevant Assays to Answer Clinical Questions

**DOI:** 10.3390/ani13193123

**Published:** 2023-10-06

**Authors:** Sophie Egyptien, Stéfan Deleuze, Joy Ledeck, Jérôme Ponthier

**Affiliations:** Equine Theriogenology, Equine Clinical Sciences Department, FARAH Comparative Veterinary Medicine, Liège University, B-4000 Liège, Belgium; segyptien@uliege.be (S.E.); s.deleuze@uliege.be (S.D.); j.ledeck@uliege.be (J.L.)

**Keywords:** stallion, horse, semen, evaluation, computer-assisted semen analysis, flow cytometry, fertility, insemination

## Abstract

**Simple Summary:**

Stallion breeding soundness is indicated for infertility or to evaluate the potential to mate or to produce and ship doses of fresh, refrigerated, or frozen semen. This review describes the methods used to evaluate equine semen and their relation to semen production pathways and fertility. Clinical examination is useful for detecting genital tract pathologies, but also for predicting the expected spermatozoa production. Basic semen evaluation in the laboratory includes analysis of the volume, spermatozoa concentration, motility, and morphology. These factors can be used to evaluate the spermatozoa production and its quality, and predict the efficiency of natural mating and the potential fresh or refrigerated semen dose production. The post-thaw quality of some stallions is below standard, despite good fresh semen quality, requiring methods for predicting this outcome. To overcome this concern and the unresolved infertility cases, biotechnologies offer new opportunities. The potential toxicity of seminal plasma can be investigated, and damaged parts of the spermatozoa can be imaged with specific staining and fluorescent microscopy or flow cytometry. These data will deepen the understanding of stallion infertility and enhance basic semen evaluation.

**Abstract:**

Stallion sperm analysis is indicated for infertility diagnosis, pre-sale expertise, production of fresh or frozen doses, and frozen straw quality control. Various collection methods are described, and numerous assays can be performed on semen. Determining an approach for each of these cases is challenging. This review aims to discuss how to obtain relevant clinical results, answering stallion owners’ concerns. Semen can be collected with an artificial vagina on a phantom or a mare, by electro-ejaculation under anesthesia, or after pharmacological induction. The collection method influences the semen volume and concentration, while the total sperm number depends on the testicular production and collection frequency. In the seminal plasma, acidity, pro-oxidant activity, and some enzymes have repercussions for the semen quality and its conservation. Moreover, non-sperm cells of seminal plasma may impact semen conservation. Motility analysis remains a core parameter, as it is associated with fresh or frozen dose fertility. Computer-assisted motility analyzers have improved repeatability, but the reproducibility between laboratories depends on the settings that are used. Morphology analysis showing spermatozoa defects is useful to understand production and maturation abnormalities. Staining of the spermatozoa is used to evaluate viability, but recent advances in flow cytometry and in fluorochromes enable an evaluation of multiple intracellular parameters. Spermatozoa protein expression already has clinical applications, for example, as a fertility and freezing ability predictor. At present, stallion semen analysis ranges from macroscopic evaluation to assessing spermatozoa proteins. However, clinically, all these data may not be relevant, and the lack of standardization may complicate their interpretation.

## 1. Introduction: What Is Your Aim?

This review aims to describe the methods of stallion semen collection and evaluation available in routine practice or in specialized labs, and their sufficiency in meeting the goals or concerns of the owner of the stallion. Semen analysis can be performed to resolve infertility cases or to assess the performance of stallions dedicated to breeding. In the latter case, the aims will vary according to the breed and the horse management. Requirements for a stallion freely mating with ten mares in a field will be less challenging than those of a thoroughbred stallion that will cover one or two mares a day, or a saddle-breed stallion collected to produce frozen semen.

Those concerns will determine all the stallion breeding soundness procedures, from the collection method to the assays performed. Collection using a phantom with a stallion only freely breeding mares could lead to inadequate conclusions about the semen volume and concentration due to difficulties mating on the dummy. Analysis of a single ejaculate of a stallion, which will be collected three times a week to produce frozen semen, is not representative of what will be required to obtain the expected number of straws. Thus, the clinical history of the stallion must also include a prospective description of its use during the next months.

Additionally, the clinical history should include a complete description of the recent mating or collections and their results, in terms of sperm quality, if data are available, or of pregnant mares, which is the final aim. Special attention should also be paid to a hypothetical anti-GnRH vaccination, as they are becoming frequently used for behavioral reasons [1]. A dedicated immune-assay of GnRH antibodies is now available and could be useful to detect vaccinated stallions [2]. Since recent pathologies could also interfere with fertility, through fever [3] or the migration of bacteria inside the male genital tract, such as during strangles [4], they should be carefully described to understand the pathogenesis and to protect the biosecurity of the facilities.

## 2. Semen Collection and Its Clinical Consequences

For stallions mating naturally, semen collection on a mare will be easier to perform because it requires less training and results in nearly real mating-condition semen. However, this collection method can be dangerous for the stallion and the operator if the mare is not correctly immobilized. Semen collection on a phantom will require more training of the stallion before it can be easily performed, and yields physiological semen. Semen can be collected using an artificial vagina (AV) on a stallion standing on all four legs in case of orthopedic pathologies, such as back pain [5]. However, this procedure can be dangerous and it is recommended to limit its use to very educated and well-mannered stallions. AVs are generally divided into two classes: those with hot water (40–42 °C) contained in a hard shell (typically, the Colorado and the INRA AVs) and those with hot water contained in a soft pocket (typically, the Missouri AV) [4,6]. The Missouri AV is lighter and easier to operate, but the temperature will decrease more quickly, whereas the Colorado AV is heavier, maintains the temperature of the water, and is sometimes preferred by stallions.

Semen collection on alert stallions provides information about libido and the semen will be more physiological, whereas electro-ejaculation under anesthesia can lead to urine contamination [7,8]. The pharmacological induction of ejaculation can be achieved using imipramine, a tricyclic antidepressant, and/or α_2_-agonsits, such as xylazine or detomidine [9,10]. The combination of both seems to be more effective, and the addition of oxytocin could help in some cases [10]. This will lead to the collection of very small volumes of semen with high spermatozoa concentrations due to the lack of seminal plasma secretion by the genital tract glands [9].

## 3. Volume and Concentration

### 3.1. Spermatozoa and Seminal Plasma Production

As in other species, Daily Sperm Output (DSO) is directly correlated to testicular volume [11]. In stallions, the following formula is used [6]: *Testicular Volume* (*TV*) (in cm^3^) = 0.5233 × *length* × *width* × *thickness* (in cm).

Then, the following formula is used to obtain the interval of the produced spermatozoa:*DSO* (×10^9^) = (0.024 × *TV*) − (0.76 to 1.26).

The testicular production is continuous, and the spermatozoa are then matured in the epididymis before being stored in the tail of the epididymis [6,12,13,14,15]. The only way to excrete the produced spermatozoa is ejaculation, which is complete after mating, whereas masturbation leads to the release of a lower number of spermatozoa [14,16]. Thus, following a sexual rest, a large proportion of ageing spermatozoa will be expelled during the first ejaculations, resulting in a large number of spermatozoa per ejaculation with a high proportion of dead spermatozoa, leading to discard the first collections’ semen [6,14]. This phenomenon is commonly referred to as the “flush period” (see Figure 1). After a long sexual rest, spermatozoa can also be agglutinated in the ampulla of ductus deferens, leading to the partial obstruction of the excretion pathways [14,17,18,19].

A stallion’s genital tract includes a prostate, as well as bulbo-urethral and vesicular glands; these are the main contributors to seminal plasma production [19]. Seminal plasma production by genital tract accessory glands is induced by sexual arousal [4,6,19,20]. Thus, a long collection time with many attempts to mate, due to inadequate training of the stallion regarding the management conditions, will result in a large ejaculate volume and lower spermatozoa concentrations. 

### 3.2. Volume, Concentration: What Are the Relevant Data?

Semen concentration is correlated with semen color, which ranges from translucid to milky aspect. However, pathological conditions can modify this color and make this appreciation inaccurate. Historically, spermatozoa concentration was manually determined with an optic microscope using counting chambers such as Thoma or Makler [6,21,22]. Thereafter, photometry methods decreased the procedure’s duration, but sometimes had weak precision [6,22]. At present, methods permeabilizing the membrane and staining of the nucleus with a propidium iodide (PI) (NucleoCounter^TM^, Chemometc, Allrod, Denmark) are reliable and useful methods for production labs [23], despite the necessary investment. Normal spermatozoa concentration in stallions varies between 100 *×* 10^6^ and 600 *×* 10^6^ spz/mL [6,24]. 

As stated above, when interpreted separately, volume and concentration values only provide insights into stallion training and collection conditions. However, when these data are used together, they can help to understand the real and observed spermatozoa production ability, using the total sperm number (TSN), which is calculated as:*TSN* (×10^6^ spz) = *Volume* (mL) × *Concentration* (×10^6^ spz/mL).

In stallions, the expected TSN is between 4 and 12 × 10^9^ spz, depending on the season, the stallion, and testes size [6,24]. After a long sexual rest, the accumulation of aging spermatozoa will lead to an increased TSN, associated with low viability and motility, from the first collection until stabilization (see Figure 1). However, the stabilization of TSN per ejaculate can lead to the recruitment of immature spermatozoa in the epididymis [14,16,25]. This can be assessed by comparing DSO and daily sperm production (DSP), calculated as follows, when the TSN is stabilized:*DSP* (over 10 days) = (*TSN_day_*_1_ + *TSN_day_*_3_ + *TSN_day_*_5_ + *TSN_day_*_8_ + *TSN_day_*_10_)/10 (=*nb.* of days).

If the DSP is higher than DSO, this could potentially mean that the flush of the aging spermatozoa stored in the tail of the epididymis is not finished yet, or that the collection program is too strenuous for the stallion, which could be confirmed by an increase in distal droplets upon spermatozoa morphological analysis [6,16,24,25]. If the DSP is lower than the DSO, an early pathology of the testis or an obstruction of the excretory ducts can be suspected, as in plugs of the ampullae [6], which can be confirmed by the decrease in alkaline phosphatase in the semen [26] and with ultrasonography [17,18].

## 4. Seminal Plasma and Non-Sperm Cells

### 4.1. General Considerations

Changes in the acido-basic status of the semen have been associated with decreased progressive motility (PM), when deviating from the ideal 7.4–7.7 pH range [4,6,22,27,28,29,30]. The optimal PM was observed for a semen osmolarity ranging from 310 to 320 mOsm [4,6,29,31]. Those parameters are still of interest, but are often forgotten in the present lab-routine semen analysis, as observed spermatozoa outcomes such as viability or motility are the most used data.

Inspection of the semen color in the lab is clinically relevant. Modifications of the color can be associated with pathologies: hemospermia, pyospermia, and urospermia will lead to red, yellow, or yellowish semen coloration, respectively. The semen’s smell will help to understand this condition, but a cytologic analysis of the semen or urea and creatinine assay will be necessary to reach the final diagnosis [4,11]. 

Hemospermia is a condition caused by traumatic, inflammatory, or tumoral lesions of the penis, the urethra, the ampulla, and the epididymis [32]. This leads to a decrease in total motility (TM) and PM [33] and can be diagnosed by the color of semen in case of large contamination or by microscopic examination [6]. Dilution of the semen with an adequate extender will preserve some motility [32], but an extended clinical examination of the whole genital tract, including ultrasonography and endoscopy, is required to detect the pathological origin of this contamination. 

Leukospermia is defined by the contamination of semen by leukocytes and is associated with acute inflammation in the genital tract [6], mainly orchitis, epididymitis, or vesiculitis. In these pathologies, neutrophils are the most commonly observed leukocytes [4,34]. These cells have deleterious effects on spermatozoa functions by producing reactive oxygen species (ROS) that are deleterious for spermatozoa [35]. In case of leukospermia, the testis and epididymis should be carefully investigated for heat or pain and an ultrasonography should be performed to confirm these conditions [4,6]. However, the most frequent inflammation localization is the vesicular glands (the other glands could also be involved [6], although this is less likely) and it can be assessed by ultrasonography and endoscopy associated with sampling through excretion ways [6].

Squamous cells are also observed in semen [36,37]. They seem to originate from accessory glands, especially the vesicular glands. No effect on fresh semen has been observed, but these cells were associated with impaired post-thaw quality by interfering with the pro-oxidant activity of the semen [37].

### 4.2. Pro and Antioxidant Activity

Redox deregulation and oxidative stress effects on spermatozoa have been investigated for many years [38,39]. Early studies investigated the antioxidant activity or the effect of antioxidant compounds and enzymes such as catalase, superoxide dismutase, reduced glutathione, and vitamin E, which improved different spermatozoa functions [39,40,41]. The origin of ROS was discussed, and two main options were proposed: the spermatozoa itself or the seminal plasma and its cellular, microbiological, or biochemical compounds.

Myeloperoxidase (MPO), a pro-oxidative enzyme of the neutrophils, was found to be correlated with post-thaw semen quality [42], but further studies failed to show a significant presence of neutrophils in fresh or frozen semen containing a large concentration of MPO [36]. Immunostaining of the semen for MPO led to localizing it in squamous non-sperm cells of the seminal plasma, and their concentrations were correlated with the MPO levels [37]. However, although MPO activity was correlated with post-thaw PM and a potential freezability marker, it was low in fresh semen, because of the partial inactivation caused by the extender [43]. 

At present, seminal plasma is thought to be the antioxidant medium, whereas the principal ROS producer is thought to be the spermatozoa, especially dead or aging ones [38,44]. However, oxidation is necessary to trigger the capacitation of the spermatozoa by allowing for cholesterol oxidation, and thus a finetuning of the equilibrium is necessary (for a high-quality review, see Pena, 2019 [38]). Commercial kits are available to measure oxidation-reduction potential (RedoxSYS^®^ Diagnostic system, Englewood, CO, USA) [45].

## 5. Spermatozoa Analysis

### 5.1. Motility Analysis

On-field semen analysis requires a 37 °C heated-plate microscope with ×1000 magnification, slides and covering slides. A visual examination of semen leads to individual subjectivity, but there are means to limit this. Motility can be evaluated in raw native semen, but samples should also be diluted with a commercial equine fresh semen extender to the same low concentration (20 to 30 × 10^6^ spz/mL) [22] to limit the natural influence of concentration on motility evaluation, as the human eyes and brain will overestimate samples with a higher concentration but the same motility [16,46,47]. Another tip would be to evaluate semen motility for only 30 s, as human eyes rapidly focus on moving spermatozoa, leading to overestimation [47]. The final piece of advice would be to train regularly and to accept that the repeatability of visual evaluations with a microscope will generally lead to variations of 10% and 5% for the more trained operators.

To address these limitations, computer-assisted semen analyzers (CASA) have been developed. Some dedicated cameras can be plugged on the heating-plate microscope, whereas other devices are designed with an inner chamber for analysis: these are more expensive but will protect the spermatozoa from external temperature and light fluctuations. Two different types of slides for analysis are also available: those where the semen drop is placed between the slide and a cover (under a cover slip on microscopic slide exam, Makler^TM^, and related devices) and those where the semen will enter the slide through the capillarity (Leja^TM^, Isos^TM^) [21]. This choice will impact the results and, thus, the setting that should be used. To obtain significant results, CASA requires an adequate concentration, usually between 20 and 30 × 10^6^ spz/mL, but dependent on the device [48], by diluting semen with a commercial extender, as previously mentioned, to analyze from 700 to 900 spermatozoa. The CASA takes pictures for 0.5 s at 60 Hz and follows the centroid of each spermatozoon to calculate the different velocities described in Figure 2 and Table 1 and Table 2. The velocity of the curvilinear path (VCL) is the velocity of the path following the 30 centroid positions acquired during the 0.5 s. The velocity of the straight-line path (VSL) is the velocity of the path between the centroid position at the beginning of the analysis and at the end of the analysis, 0.5 s later. The velocity average path (VAP) uses a geometrical smoothing of the VCL on the VSL and determines the velocity of this path over the period of 0.5 s. The computer will also determine the linearity (LIN), straightness (STR), and wobbling (WOB) using these parameter ratios (see Table 1).

Once the raw parameters are acquired by the CASA, it will determine TM and PM using the settings defined by the user, mainly depending on the type of slides that are used. In Table 2, the main internationally recognized settings are listed: in stallions, the main raw parameters used to determine TM and PM are VAP and STR [4,6,16,47]. The values of TM and PM remain difficult to compare between laboratories, despite standardization efforts. Total motility should be above 70% and PM should be above 50–60% for the raw semen of a normal stallion [4,16,47]. However, the total number of progressively motile spermatozoa in the dose is becoming the main fertility-related factor in fresh and frozen semen [6,47]. 

Other systems using light absorbance have been developed to analyze motility in semen, but at first suffered from a lack of repeatability [21]. At present, portable devices are being developed and some have shown repeatable results that correlate with those achieved by conventional CASA devices [49].

### 5.2. Morphology Evaluation and Clinical Outcomes

The most frequent technique used to evaluate spermatozoa morphology is to smear semen on a slide, air-dry it, stain it using different available means of staining (Wright, Giemsa, hematoxylin-eosin), and observe at least 100 sperm cells with a common microscope at ×1000 magnification [22]. Diff-Quick commercial kits are designed to provide a mean of quick, easy, and on-field staining for semen [22,36,50], which will give practitioners access to basic morphological evaluations that could be clinically relevant. However, this technique may lead to some damage to the spermatozoa, such as separated heads, which will induce misinterpretation [25], and the staining can impair the visualization of small structural details [22]. Lengthening the Diff-Quick staining timings to 5 or 30 min [51,52] can lead to an observed slide quality that is similar to the reference method, but impairs its main interests: the speed and ease of realization. At present, the reference technique for fine morphology analysis is to dilute semen (1v/40v) in 37 °C formaldehyde 4%, to place a 5 µL drop on a 37 °C slide under a covering slide, and then to analyze 200 spermatozoa with contrast microscopy at ×1000 [6,25]. Dedicated morphology applications are included in some CASA devices. According to the author’s experience, this feature can be useful in everyday production [53], but should be regularly controlled by a manual morphology evaluation, as some abnormalities are not properly assessed by the CASA. 

Morphological abnormalities of spermatozoa are divided into primary abnormalities, related to defects in testicular production, secondary abnormalities, related to impaired spermatozoa maturation in the excurrent duct system, and tertiary abnormalities resulting from improper semen collection or handling procedures [22]. As an example, the distal droplet increase in semen is related to overuse of the stallion, leading to the excretion of spermatozoa that are not totally matured in the epididymis, with a DSP that is larger than the DSO [6,22,25,54]. A cold shock on the slide will also lead to abnormalities of the extremity of the final piece, such as a rolled tail [25]. Multiple or enlarged heads or swollen intermediary pieces are typically related to abnormalities in the production in the testis [25]. Moreover, amorphous heads or heterogenous nuclei have been associated with capacitation or acrosome reactions and DNA fragmentation, respectively [54,55], which are more related to secondary abnormalities caused by seminal plasma composition or semen conservation procedures. The Sperm Deformity Index (SDI) is a computer-assisted microscopic analysis of the spermatozoon nucleus morphology that has been associated with stress-induced DNA damages [56], but requires specific microscopic equipment.

The overall proportion of abnormalities should not exceed 35% in stallions, with a maximum of 5% and 10% for each primary and secondary abnormality, respectively [6,25]. However, a single-time value will not provide useful information to help to understand the process’ evolution. As a first example, the high proportion of distal droplets shows that production is impeded, but a decrease in this abnormality in further semen collections means that management has been improved. Another example is an increase in a primary abnormality, such as swollen intermediary pieces, from 3 to 6%: this is a real signal of ongoing testicular degeneration. These considerations highlight the importance of semen evaluation on several different spermograms to assess the processes’ evolution.

### 5.3. On-Field Viability Analysis

In the current andrology vocabulary, viability is defined by the World Health Organization as the membrane integrity, although the mitochondrial activity or DNA integrity could also be considered viability factors. Eosin–nigrosin staining is an easy on-field evaluation of the membrane, as eosin only crosses the membranes of dead spermatozoa [6], whereas alive spermatozoa remain unstained. Using the same logic, devices used to assess concentration with the propidium iodide staining of the spermatozoa (NucleoCounter^TM^, Chemometc, Allrod, Denmark) can be used to assay the proportion of intact membrane spermatozoa when the membrane is not permeabilized before the assay [23]. These methods are easy to perform in the lab and can avoid the use of flow cytometry. Viability obtained with this definition is normally over 70% and normally quite well associated with TM.

## 6. Fluorescent Microscopy and Flow Cytometry

Fluorochromes are markers of specific parts of the sperm cell that emit light of a known wavelength when excited by specific lasers. The emitted light is directly observed under fluorescent microscopy or recorded by flow cytometry. The flow cytometry technique analyzes one cell at a time, as it passes one cell after another in front of the excitation laser. Each emission answer is recorded and plotted as an event. Multiple parameter analyses are possible using different fluorochromes, avoiding spectral overlap. This could improve the robustness of semen evaluation. Classic fluorescent microscopy is manual; the number of evaluated events is limited by different parameters including photo bleaching and semen motility. Flow cytometry evaluates thousands of events in minutes and intrinsically avoids fluorescent microscopy’s limitations. Imaging flow cytometry uses a flow cytometer with a microscope combined with a camera to observe the morphology of one spermatozoon at a time. Some CASAs are associated with fluorescent microscopes, allowing for the simultaneous evaluation of morphology and mobility, as well as some functional evaluations. However, interestingly, this tool is not generally available and flow cytometry is far more widely used. It has recently been used in stallions to study the association of morphological abnormalities (head, proximal droplets, midpiece, and coiled tail) with ROS production [57]. Generally, flow cytometry evaluates viability, acrosome status, mitochondrial potential, DNA fragmentation, and ROS production. 

The live/dead viability kit for semen, commercialized by ThermoFisher^®^ (Merelbeke, Belgium), uses SYBR and propidium iodide (PI) fluorochromes, which are DNA-binding fluorochromes [58,59]. SYBR, as a membrane permeable molecule, marks all nuclei in green, and PI, a membrane impermeable fluorochrome, marks cell nuclei with impaired membrane integrity by quenching SYBR [58]. This is the most classical fluorescent viability kit and it is used in several domestic species to evaluate semen viability. Its limitations are its incompatibility with cell fixation and its consumption of two channels to analyze one parameter, which restricts its use in multiple-parameter analyses. Recently, calcein violet fluorochrome has also been validated as a semen membrane integrity assay in different species, including horses [60]. The hydrolyzation of its permeant non-fluorescent form into an impermeable blue fluorescent form is detected in cells with intact membrane integrity. Using these kits in flow cytometry will enable the easy assessment of viability for the largest number of events that an operator could deal with [60].

Another core parameter is the evaluation of the acrosomal membrane integrity. Arachis hypogea agglutinin is a lectin that binds to β-galactose moieties present on the inner part of the outer membrane of the acrosome membrane. Another lectin, Pisum sativum agglutinin, although less specific to the acrosome membrane than the peanut equivalent, has been widely used to evaluate acrosomal status [39]. This lectin mainly binds to the acrosomal matrix and, to a lesser extent, to the cytoplasmic membrane matrix [61]. They both may be labelled with fluorochromes fluorescein isothiocyanate (FITC) or phycoerythrin (PE) [62]. FITC emits a bright green fluorescence and is often preferred to PE, which emits in the light red-orange spectrum but suffers severely from photobleaching, limiting its efficiency in fluorescent microscopy [62]. These labelled lectins are membrane-impermeable; marking the acrosome without prior permeabilization indicates an acrosomal reaction [62]. In the absence of permeabilization, cell membrane integrity and viability should be simultaneously evaluated to discriminate the sperm cells that lost their membrane integrity from those with a reacted acrosome [62]. A functional alternative to assay capacitation and acrosomal integrity is the zona-binding assay, which evaluates the binding of spermatozoa to the separated zona pellucida of either homologous or heterogenous (bovine) oocytes [63].

Furthermore, ROS production (dichlorodihydrofluorescein diacetate for H_2_O_2_, dihydroethidium for superoxide anion, and CellROX deep red for total ROS), as well as the effect of ROS on lipid peroxidation (BODIPY-C11), can be assessed using flow cytometry [64]. Among the domestic animals, the equine spermatozoa have the greatest mitochondrial activity and, thus, the greatest ROS production [38]. MitoTracker Deep Red (Life Technologies^®^_,_ ThermoFisher^®^_,_ Merelbeke, Belgium) is a validated option that may be used in fixed samples [65]. Two obvious advantages are its fixability, allowing for the evaluation of semen hours later, as opposed to needing to rush to the lab just after semen collection. The second one is its wavelength: the deep red will not overlap with other fluorochromes used to evaluate different parameters in a panel. In this validation study [65], it was used in combination with live/dead Zombie Green^®^ (Biolegend, Amsterdam, The Netherlands), another membrane-impermeable fixable dye that fixates on amines from the membrane and the cytoplasm. This results in a higher marking of cells with impaired cell membranes than those with an intact membrane. These kinds of zombie dyes come in different emission wavelengths and their use will most probably overtake the classical SYBR/PI viability dye in future. Assessing mitochondrial activity remains of interest, as recent studies highlighted the association between this parameter and fertility (for a high-quality review, see Pena, 2019 [38]).

An indirect way of evaluating viability is the evaluation of apoptosis. The test uses the following non-permeable DNA-binding molecules: Yo-Pro-1 and ethidium homodimer-1. Yo-Pro-positive (green fluorescence) cells are considered in the early stage of apoptosis: those positive for ethidium homodimer are considered necrotic and double-positive cells are in the late apoptosis stage. Negative cells are alive, without membrane integrity alterations [66]. The exclusion of negative non-cellular events may be needed; this can be achieved using Hoechts33342, a membrane-permeant nucleic blue stain [67]. Sperm chromatin denaturation is related to the level of DNA strand breaks, which can be caused by factors such as idiopathic apoptosis, oxidative stress, and radiations exposure [22]. Sperm chromatin structure can be assayed using flow cytometry. The main interest lies, for instance, in reported cases of low fertility but with a spermogram that seems normal in terms of motility and viability [68]. Acridine orange (AO), a metachromatic dye, evaluates the DNA fragmentation index. This is the ratio of single- and double-stranded DNA present in sperm cells stressed by an acidic environment. Fluorescence is emitted by cells with a single-stranded shift from green to light red-orange wavelengths. Sperm chromatin dispersion relies on the halo formed by intact DNA loops in spermatozoa after nuclear proteins’ denaturation and extraction [69], and is validated in stallions [70,71]. However, these assays are based on the observation of halos on agar slides [69], which makes them unsuitable for flow cytometry analysis, leading to inter-operator subjectivity. However, new computer-assisted methods are being developed to overcome this limitation [69].

Easy-to-use and semen-dedicated flow cytometry devices are available in routine semen-production laboratories, leading to these technologies being used in common practice. However, the results depend on the settings used in each lab and at the beginning of each analysis: therefore, they should only be compared with the data observed during the same set of analyses.

## 7. Proteomics

At present, intracytoplasmic protein markers and assays are used to evaluate both fresh and post-thaw semen quality. The concentration of proAKAP4, the main structural protein of the fibrous sheath of spermatozoa [72], has been correlated with post-thaw quality in bulls [73] and horses [74]. Confirmation of these data could make of proAKAP4 a useful tool in semen-freezing facilities [72,74]. A recent large-scale analysis of the spermatozoa proteome in fresh semen showed the differences between good and bad freezer samples [75], and the identified proteins were linked to the redox status [44,45]. A proteomic evaluation of the stallion semen seems to be the next technology needed to predict its freezability. However, easy and reliable devices should be developed to ensure soundness of breeding and the semen production units. 

## 8. Semen Microbiological Quality

Semen collection is a non-sterile procedure, except when it is performed by the aspiration of the epididymis. Seminal plasma is a perfect culture medium for bacteria, and equine semen is known to contain a variety of bacterial species. Among these, some have the potential for venereal transmission in the equine, including *Taylorella equigenitalis, Klebsiella pneumoniae*, *Pseudomonas aeruginosa,* and possibly beta-hemolytic streptocci such as *Streptococcus equi equi* [30], thus leading to the use of specific stallion and semen management practices. However, many other bacteria are non-pathogenic, such as various species of *Streptococcus, Acinobacter*, *Aerobacter*, *Proteus*, *Staphylococcus*, and *E. coli* [30]. Recent research suggests that these bacteria may have an impact on both fresh and cold semen conservation [64,76,77,78]. Consequently, methods to control their growth in semen, without adding empirically broad-spectrum antibiotics to the extender, are currently investigated [76,78].

## 9. Conclusions: How Can the Data Set Be Summarized with Relevant Clinical Information? 

All the devices and analyses described above will provide information to the veterinary clinician or the semen production operator, and such large amounts of data can be confusing. However, a comprehensive clinical strategy will enable the choice of the right diagnosis pathway. At present, progressive motility and the total number of progressive spermatozoa per dose are the only factors that have been strongly and directly associated with observed fertility in mares [46,47,79,80]. Viability is somehow associated with the progressive motility, but other data, such as morphology or fluorescent staining, used with either microscopy or cytometry, will allow for an understanding of the origin and could be used to propose treatments for the semen in the lab or for the stallion. 

As stated above, defining the aim of the stallion will allow for thresholds to be established for different uses. For natural mating, fresh, or refrigerated semen, it is generally accepted that the best fertility is observed with 300 to 500 × 10^6^ progressively motile spermatozoa per dose [6,80]. Clinical examination of the stallion and basic repeated semen analysis including volume, concentration, motility, morphology, as well as a viability analysis, will allow for an evaluation of what will be possible for the stallion. In the case of fresh and refrigerated semen, the concerns regarding semen quality during shipping should be addressed. If the progressive motility is significantly decreasing, the total number of progressive spermatozoa could be adjusted during production, but the toxicity of seminal plasma should be investigated. Hemospermia, leukospermia, urospermia, or ROS could explain this toxicity [6,7]. In the case of frozen–thawed semen, the main concern is a bad post-thaw motility, despite a good fresh semen quality [36,37,47]. Previous and current studies [37,43,44,45,74,75] are focused on the development of assays to predict post-thaw quality, but different extenders and methods of freezing [37,81] should be tested to improve the frozen semen quality of these “poor-freezers”.

In some cases, although the recommended number of progressive spermatozoa per dose is used, breeding or insemination with fresh, refrigerated, or frozen semen does not lead to sufficient pregnancy rates. In these cases, technologies such as flow cytometry, proteomic analysis, and other biotechnologies are of interest for assessing DNA integrity, ROS, and mitochondrial activity, or other potential factors. However, the required devices and competences are expensive and their results are not standardized. Basic clinical and semen examination is still necessary, but new technologies offer a wide range of exciting assays, as long as one does not drown under the volume of data and maintains a progressive, clear, and comprehensive strategy.

## Figures and Tables

**Figure 1 animals-13-03123-f001:**
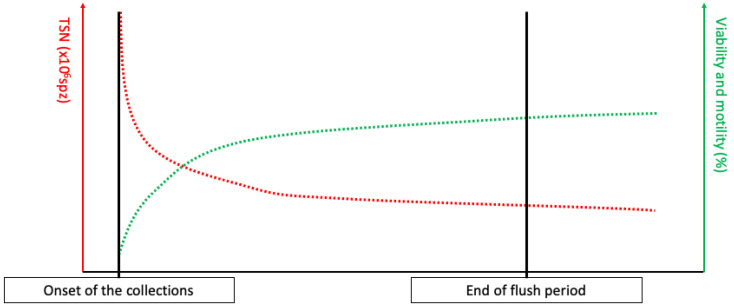
Total sperm number (TSN), viability, and motility evolution during the flush period, defined as the period of collection necessary to empty the ageing spermatozoa in the tail of epididymis.

**Figure 2 animals-13-03123-f002:**
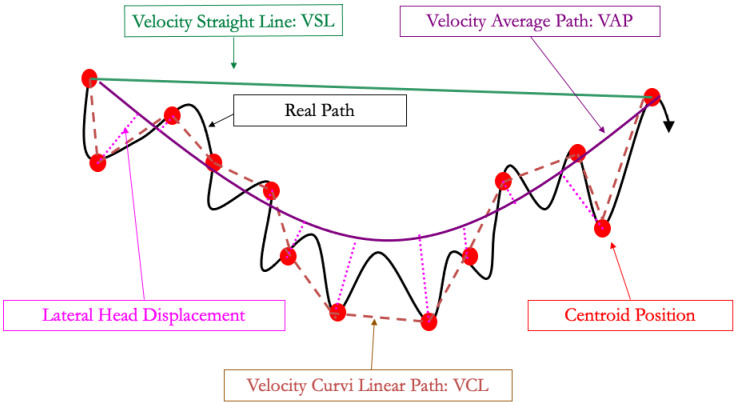
Spermatozoa pathways used in computer-assisted semen analysis.

**Table 1 animals-13-03123-t001:** Computer-assisted semen analyzer (CASA) parameters.

Abbreviation	Name	Definition
VCL	Velocity curvilinear path	Velocity of path following centroid position during analysis.
VSL	Velocity straight-line path	Velocity of path between beginning and end of centroid position during analysis.
VAP	Velocity Average Path	Velocity of the smoothed VCL.
LIN	Linearity	VSL/VCL.
STR	Straightness	VSL/VAP.
WOB	Wobbling	VAP/VCL.

**Table 2 animals-13-03123-t002:** Computer-assisted semen analyzer (CASA) settings for different types of slides.

	Drop	Capillarity
Total Motility	VAP > 10–15 µm/s [13] VAP > 20 µm/s [47]	VAP > 15 µm/s [6]
Progressive Motility	VAP > 10–15 µm/s and STR > 100% [13] VAP > 40 µm/s and STR > 80% [47]	VAP > 30 µm/s and STR > 50% [6]

## Data Availability

All references of this bibliographical are available in the reference section.

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
