# Peer review of "Sperm Quality Assessment in Stallions: How to Choose Relevant Assays to Answer Clinical Questions"

_animals, 2023, doi:10.3390/ani13193123_

Round 1

Reviewer 1 Report

Overall, the manuscript “Sperm Quality Assessment in Stallions: How to Choose 2 Relevant Assays to Answer Clinical Questions” brings exciting information for practitioners in the equine industry. However, the manuscript lacks details and can be improved to clarify the information. The authors must better organize these sections before publication.

The simple summary is confusion, and it would be better if the authors can clarify the objectives of this review.

Abstract

L26 - relevant clinical results for what? Please, complete this sentece

L27 - The authors describe electroejaculation as a method of semen collection in stallions but do not describe other much more common methods, such as chemical ejaculation. Please review.

L36 - "analysis"???

Introduction

The introduction lacks in objectives for the review and it's a bit confusion at some points. 

L 72 - Artificial insemination is not the first time the appears, please provide the abbreviation at the first appearance.

Please, expand chemical ejaculation in the section "Semen Collection and Its Clinical Consequence"

L92 - What do you mean by "incomplete after masturbation?

L97 - 100 - Please, expand and clarify this sentence.

Figure 1. - Describe what mean TSN in the Figure. What does "flush period" mean?

L 127 - 132 - Please, clarify this sentence.

L205 - after [17] needs a period "."

L234 - Morphology Evaluation and Clinical Outcomes - It's missing techniques, please describe the techniques available.

Expand the Diff Quick techniques - Ex.:

Pozor. Usefulness of Dip Quick Stain in evaluating sperm morphology in stallions.

Segabinazzi et al. Dip Quick Staining Modified for Morphological Evaluation to Equine Spermatozoa

L 279 - Fluorescent Microscopy and Flow Cytometry - Split the assays and expand fluorescent microscopy

L358 - must be expanded

Author Response

Reviewer 1

Overall, the manuscript “Sperm Quality Assessment in Stallions: How to Choose 2 Relevant Assays to Answer Clinical Questions” brings exciting information for practitioners in the equine industry. However, the manuscript lacks details and can be improved to clarify the information. The authors must better organize these sections before publication.

All the comments were used to clarify the information and the manuscript.

The simple summary is confusion, and it would be better if the authors can clarify the objectives of this review.

Thank you for your suggestion, we have totally rewritten the simple summary in that way.

Abstract

L26 - relevant clinical results for what? Please, complete this sentence

Thank you for this relevant comment, this sentence has been completed.

L27 - The authors describe electroejaculation as a method of semen collection in stallions but do not describe other much more common methods, such as chemical ejaculation. Please review.

This method has been added.

L36 - "analysis"???

It has been modified.

Introduction

The introduction lacks in objectives for the review and it's a bit confusion at some points.

Thank you for this remark. In response, we have introduced clear aims in the beginning of the paragraph, and have also rephrased some sentences to enhance clarity and comprehension.

L 72 - Artificial insemination is not the first time the appears, please provide the abbreviation at the first appearance.

Thank you for this comment, it has been modified in the manuscript.

Please, expand chemical ejaculation in the section “Semen Collection and Its Clinical Consequence”

We have added an additional sentence to describe consequences of employing this technique.

L92 - What do you mean by "incomplete after masturbation?

We have rephrased this sentence.

L97 - 100 - Please, expand and clarify this sentence.

This sentence has been re-formulated.

Figure 1. - Describe what mean TSN in the Figure. What does "flush period" mean?

Thank you for your comment, it has been modified in the manuscript.

L 127 - 132 - Please, clarify this sentence.

The sentence has been modified to improve reader’s comprehension.

L205 - after [17] needs a period "."

Thank you for your careful review: it has been modified.

L234 - Morphology Evaluation and Clinical Outcomes - It's missing techniques, please describe the techniques available.

Expand the Diff Quick techniques - Ex.:

Pozor. Usefulness of Dip Quick Stain in evaluating sperm morphology in stallions.

Segabinazzi et al. Dip Quick Staining Modified for Morphological Evaluation to Equine Spermatozoa

Our recommendation regarding contrast microscopy has been more precisely and detailed argued.

L 279 - Fluorescent Microscopy and Flow Cytometry - Split the assays and expand fluorescent microscopy

To enhance reader comprehension, we believe that an approach by part of the spermatozoa is easier to understand for the reader who may not possess specialized knowledge and skills about these specialized lab devices. Moreover, separating the sections on flow cytometry and fluorescent microscopy would lead to repetitions and increase the length of this manuscript.

L358 - must be expanded

We re-formulated this sentence.

Reviewer 2 Report

Dear authors,

thanks a lot for writing a review which addresses the important topic on spermatological examination of stallion ejaculates. The review is well-written and gives an overview on the state of spermatological examination in the horse. However, the manuscript would benefit from including several additional tools of semen analysis as determination of redox-state, halo assay and Zona-binding assay.

A list of specific comments can be found here:

. 12 „expected“

l. 23 „straw“

l. 37 please correct the use of „proteomic“

l. 49-51 please provide a reference for this hypothesis or use another example here

l. 69-72 please consider citing Burger et al. 2015 here (https://doi.org/10.1016/j.theriogenology.2015.04.029)

l. 90-92 please mention that semen might also be stuck in the ampullae

l. 83-100 please refer to the “flush period” since it is mentioned in Fig. 1

l. 131 or 132 besides ASP measurement the ultrasonographic examination of the genital tract may also provide useful information

l. 136 Why don’t use a range here as given by other authors? For some stallions, lower pH (e.g., pH 7.2) might be physiological, too.

l. 150-151 please specify the meaning of “extended clinical examination”

l. 186 magnification of microscope is missing

l. 188-197 motility analysis should not be restricted to extended semen samples, but also be performed in native semen samples

l. 204 What is the meaning of “…”?

l.232-233 abbreviations should be avoided in table legends.

l. 242-246 According to Varner (2008) and others, there is a third category: tertiary defects. Please consider using and explaining three different categories here.

l. 345-346 Please use a citation here.

l. 345-352 please also include the halo assay for determination of DNA fragmentation in your review

Information on the importance on genital microbial health on semen quality should be provided in the review, too.

Author Response

Reviewer 1

Dear authors,

thanks a lot for writing a review which addresses the important topic on spermatological examination of stallion ejaculates. The review is well-written and gives an overview on the state of spermatological examination in the horse.

However, the manuscript would benefit from including several additional tools of semen analysis as determination of redox-state, halo assay and Zona-binding assay.

Redox measurement has been added (line 196) and redox effect assay by cytometry has been added (line 343).

 Halo assay has been added (line 372).

Zona binding assay has been added as a functional alternative to flow-cytometry evaluation of acrosome (line 340).

A list of specific comments can be found here:

  1. 12 „expected“

Thank you: it has been modified.

  1. 23 „straw“

As we use more than one straw for quality control, and unpublished data from our colleagues’ team seems to prove that they can be quality variations within the same batch, we prefer to keep the plural form.

  1. 37 please correct the use of „proteomic“

It has been re-formulated.

  1. 49-51 please provide a reference for this hypothesis or use another example here

This sentence has been modified to address both your remark and that of the first reviewer.

  1. 69-72 please consider citing Burger et al. 2015 here (https://doi.org/10.1016/j.theriogenology.2015.04.029)

This reference has been added.

  1. 90-92 please mention that semen might also be stuck in the ampullae

Thank you for that meaningful remark, it has been added.

  1. 83-100 please refer to the “flush period” since it is mentioned in Fig. 1

This has been re-formulated and defined in the text.

  1. 131 or 132 besides ASP measurement the ultrasonographic examination of the genital tract may also provide useful information

This has been added in the manuscript with the references.

  1. 136 Why don’t use a range here as given by other authors? For some stallions, lower pH (e.g., pH 7.2) might be physiological, too.

The range has been added in the manuscript.

  1. 150-151 please specify the meaning of “extended clinical examination”

We addressed this comment in manuscript.

  1. 186 magnification of microscope is missing

It has been added.

  1. 188-197 motility analysis should not be restricted to extended semen samples, but also be performed in native semen samples

This has been mentioned in the manuscript.

  1. 204 What is the meaning of “…”?

We specified that it was the same kind of devices.

l.232-233 abbreviations should be avoided in table legends.

We modified the titles using Computer Assisted Semen Analysis rather than CASA. However, we kept the abbreviations as VAP, VSL, VCL, STR (…) as they are used in devices and useful for the readers.

  1. 242-246 According to Varner (2008) and others, there is a third category: tertiary defects. Please consider using and explaining three different categories here.

Thank you for that remark: we modified the manuscript, and it really improves it.

  1. 345-346 Please use a citation here.

We modified the text to be less assertive and added a reference.

  1. 345-352 please also include the halo assay for determination of DNA fragmentation in your review

We added 2 sentences in the paragraph dealing with DNA integrity analysis.

Information on the importance on genital microbial health on semen quality should be provided in the review, too.

We added a last paragraph on this subject as it is of interest for practitioners, but maybe not in the scope of this review.

Round 2

Reviewer 1 Report

The paper has been improved after the first review. However, the authors ignored some of the pointed concerns. This paper is a "review"; therefore, all the information found in the literature should be described.

Please include a brief description of the protocols for chemical ejaculation.

In addition, the authors ignored "L234 - Morphology Evaluation and Clinical Outcomes - It's missing techniques; please describe the techniques available for practitioners." Diff Quick is the most used staining technique in clinical practice, and it should be expanded - Why is this staining not adequate for sperm analysis, and what strategies can improve its use?! Please review and expand.

Please connect the tests and results with fertility rates.

Author Response

Dear Doctor Papas,

Dear Doctor Catalan,

Once again, we would like to thank you and the reviewers for their contributions that improved this manuscript.

You will find below the answers to the reviewer 1 questions, comments and corrections. All required changes have been highlighted in blue in the new manuscript, as in the answers to the comments.

I hope they will match with your journal standards and with reviewers’ recommendation.

Sincerely yours,

Dr. Jérôme PONTHIER

DVM, M. Sc., Ph. D., Diplomate ECAR

Reviewer 1

The paper has been improved after the first review. However, the authors ignored some of the pointed concerns. This paper is a "review"; therefore, all the information found in the literature should be described.

Please include a brief description of the protocols for chemical ejaculation.

This has been added (line 85 to 88).

In addition, the authors ignored "L234 - Morphology Evaluation and Clinical Outcomes - It's missing techniques; please describe the techniques available for practitioners." Diff Quick is the most used staining technique in clinical practice, and it should be expanded - Why is this staining not adequate for sperm analysis, and what strategies can improve its use?! Please review and expand.

As the modifications made in our V2 manuscript were not sufficient, we modified it again in the v3 manuscript. We hope that this will match with your recommendations. (Line 253-261)

Please connect the tests and results with fertility rates.

We added some comments on this concerns in the conclusions part (Line 423-428).

Round 3

Reviewer 1 Report

The authors did a great job in this last version of the manuscript, which improved its quality.

I would like to ask the authors to include the approaches that can be used in clinical practice to improve the Diff Quick technique semen analysis. Please, check and include:

Pozor. Usefulness of Dip Quick Stain in evaluating sperm morphology in stallions.

Segabinazzi et al. Dip Quick Staining Modified for Morphological Evaluation to Equine Spermatozoa

After that, the paper can be published and will help practitioners in the field to select the best approach for stallion semen evaluation.

Author Response

Dear Reviewer,

Once again, We would like to thank you for the amount of work. 

We added a sentence about your remark (lines 261-263, in violet).

Best,